# Improving Electrochemical Performance at Graphite Negative Electrodes in Concentrated Electrolyte Solutions by Addition of 1,2-Dichloroethane

**Hee-Youb Song [1], Moon-Hyung Jung [2] and Soon-Ki Jeong [1],***  

[1]   Department of Energy Systems, Soonchunhyang University, Asan, Chungnam 31538,
     Korea; youbi0815@sch.ac.kr
[2]   VITZROCELL Co., Ltd., 70, Indeoseupakeu, Hapdeok, Dangjin, Chungnam 31816, Korea; blueboil@sch.ac.kr
*   Correspondence: hamin611@sch.ac.kr

**Abstract:** In concentrated propylene carbonate (PC)-based electrolyte solutions, reversible lithium intercalation and de-intercalation occur at graphite negative electrodes because of the low solvation number. However, concentrated electrolyte solutions have low ionic conductivity due to their high viscosity, which leads to poor electrochemical performance in lithium-ion batteries. Therefore, we investigated the effect of the addition of 1,2-dichloroethane (DCE), a co-solvent with low electron-donating ability, on the electrochemical properties of graphite in a concentrated PC-based electrolyte solution. An effective solid electrolyte interphase (SEI) was formed, and lithium intercalation into graphite occurred in the concentrated PC-based electrolyte solutions containing various amounts of DCE, while the reversible capacity improved. Raman spectroscopy results confirmed that the solvation structure of the lithium ions, which allows for effective SEI formation, was maintained despite the decrease in the total molality of $LiPF_6$ by the addition of DCE. These results suggest that the addition of a co-solvent with low electron-donating ability is an effective strategy for improving the electrochemical performance in concentrated electrolyte solutions.

**Keywords:** solid electrolyte interphase; lithium-graphite intercalation compounds; concentrated electrolyte solutions; lithium-ion batteries; solvation structures

## 1. Introduction

Electrochemical lithium intercalation into graphite negative electrodes is a significant issue when considering the diversity of electrolyte solutions applicable to lithium-ion batteries (LIBs) [1–4]. Lithium intercalation occurs when the electrode (graphite) is covered with a surface film called the solid electrolyte interphase (SEI), whose formation is strongly dependent on the kind of electrolyte species. Ethylene carbonate (EC)-based electrolyte solutions allow for the formation of an effective SEI during the initial charge process. Hence, EC is most widely used as the main solvent in LIBs. In contrast, electrolyte solutions based on propylene carbonate (PC) and ether (e.g., tetrahydrofuran, glyme-based solvents) lead to continuous solvent co-intercalation and electrolyte decomposition instead of SEI formation and lithium intercalation [5–7].

Interestingly, the electrolyte solutions with high salt concentration are conducive to SEI formation, and thus facilitate lithium intercalation into graphite negative electrodes [8–11]. In this case, the solvation structure of the lithium ion drastically changes because of the decrease in the solvation number as compared with that in low-concentration electrolyte solutions. Since solvation structure is a critical factor influencing SEI formation on graphite negative electrodes, various strategies for controlling the solvation structure have been proposed (e.g., increasing the salt concentration and the

addition of a Lewis acid or base) [8–14]. Among these, increasing the salt concentration is a simple method for forming an effective SEI at graphite negative electrodes. However, salt addition largely increases the viscosity of the electrolyte solution resulting from forming ion pairs and aggregates, consequently decreasing the ionic conductivity, and thereby degrading the electrochemical performance of LIBs [15]. Therefore, a method that improves the ionic conductivity is needed to enhance the electrochemical performance of LIBs in concentrated electrolyte solutions.

In this study, we focused on concentrated PC-based solutions. This was because PC has a lower melting point ($-49$ °C) compared to EC (34 °C) and thus has excellent ion conductivity at low temperatures. We investigated the effect of the addition of 1,2-dichloroethane (DCE) on the electrochemical performance of graphite in concentrated PC-based electrolyte solutions. Because of the low electron-donating ability of DCE (donor number: 0), the solvation structure that is instrumental for effective SEI formation was expected to be maintained along with a decrease in the electrolyte viscosity and an increase in the ionic conductivity [16]. To the best of our knowledge, this is the first report on a method for improving ionic conductivity while maintaining the solvation structure of lithium ions in concentrated electrolyte solutions. The electrochemical properties were investigated in the concentrated PC-based electrolyte solutions containing various amounts of DCE. The structures of the electrolyte solutions were clarified by Raman spectroscopy.

## 2. Materials and Methods

The concentrated PC-based electrolyte solution was prepared by dissolving $LiPF_6$ (battery grade, Enchem Ltd., Chungbuk, Korea) in PC (molar ratio of PC/$Li^+$ = 3:1). PC (battery grade, Enchem Ltd., Chungbuk, Korea) and DCE (anhydrous, 99.8%, Sigma-Aldrich Co., St. Louis, USA) mixed electrolyte solutions were prepared by adding DCE as a co-solvent to the concentrated PC-based electrolyte in various molar ratios (PC/$Li^+$/DCE = 3:1:0, 6:2:1, 3:1:1, and 3:1:2). Hereafter, the prepared electrolyte solutions will be denoted as PLD310, PLD621, PLD311, and PLD312 according to the molar ratio of PC, $Li^+$, and DCE.

Natural graphite (CGB-10, Nippon Graphite Industries, Ltd., Tokyo, Japan) was used as the working electrode. Natural graphite powder and poly(vinylidene fluoride) (PVdF, average MW 534,000, Sigma-Aldrich Co., St. Louis, USA) binder were mixed in 1-Methyl-2-pyrrolidinone (graphite/PVdF = 9:1, wt%) for preparing a slurry, which was then coated on Cu foil and dried in a vacuum oven at 120 °C for 12 h. Lithium foil was used as the counter and reference electrodes. A three-electrode cell was used for investigating the electrochemical properties in the prepared electrolyte solutions. Cyclic voltammetry (CV, ZIVE MP2A, WonATech Co., Ltd., Seoul, Korea) tests were carried out at 0.5 mV s$^{-1}$, and charge/discharge tests were performed at a constant current of 0.5 C (1 C = 372 mA g$^{-1}$), between 2.7 and 0.0 V.

Raman spectra were obtained by using a 633 nm HeNe ion laser (Nanofinder 30, Tokyo Instruments, Inc., Tokyo, Japan) to investigate the solvation structures in the electrolyte solutions.

## 3. Results and Discussion

Table 1 shows the physical properties of the PC-based electrolyte solutions. The viscosity of concentrated electrolyte solutions is much higher than that of normal electrolyte solutions because of the stronger interactions among the cations, anions, and solvent molecules, leading to the formation of ion pairs and aggregates [17]. It follows that the aforementioned interactions should be rendered weaker for decreasing the viscosity; this in turn increases the ionic conductivity in concentrated electrolyte solutions. PC-based electrolyte solutions containing various amounts of DCE showed higher ionic conductivity than did the concentrated PC-based electrolyte solution without DCE because of the low viscosity. Moreover, the ionic conductivity increased with increasing amounts of DCE. These results indicate that the addition of DCE effectively decreased the viscosity due to the weak interaction among the ions and solvent molecules in the concentrated PC-based electrolyte solution.

**Table 1.** Physical properties of electrolyte solutions. DCE: 1,2-dichloroethane; PC: propylene carbonate.

| Electrolyte Solution | Concentration (mol kg$^{-1}$) | Molar Ratio (PC/LiPF$_6$/DCE) | Viscosity (Pa s) | Ionic Conductivity (mS cm$^{-1}$) |
| --- | --- | --- | --- | --- |
| PLD310 | 3.27 | 3:1:0 | 0.580 | 0.41 |
| PLD621 | 2.81 | 6:2:1 | 0.117 | 0.85 |
| PLD311 | 2.47 | 3:1:1 | 0.059 | 1.31 |
| PLD312 | 1.98 | 3:1:2 | 0.030 | 2.84 |

In order to evaluate the electrochemical properties, CV was carried out at the graphite negative electrode in the concentrated PC-based electrolyte solutions containing various amounts of DCE (Figure 1). In all cases, reversible reduction and oxidation peaks were confirmed under 0.5 V, which could be attributed to lithium intercalation and de-intercalation into/from graphite. Moreover, a cathodic current was observed above 0.5 V during the first cycle, which decreased after the first cycle. This result revealed that above 0.5 V, the electrolyte solution decomposed to form an SEI. It can also be considered that an effective SEI was formed in the PC and DCE mixed electrolyte because of the reversible lithium intercalation and de-intercalation reactions. In other words, the solvation structure of the lithium ions, which enables the formation of an effective SEI, was maintained even though the total molality of LiPF$_6$ was decreased with the addition of DCE in the concentrated PC-based electrolyte solutions. The cathodic and anodic currents corresponding to lithium intercalation and de-intercalation also increased with increasing amounts of DCE; that is, the electrochemical performance improved when the ionic conductivity of the electrolyte solutions, which was instrumental in SEI formation, was increased.

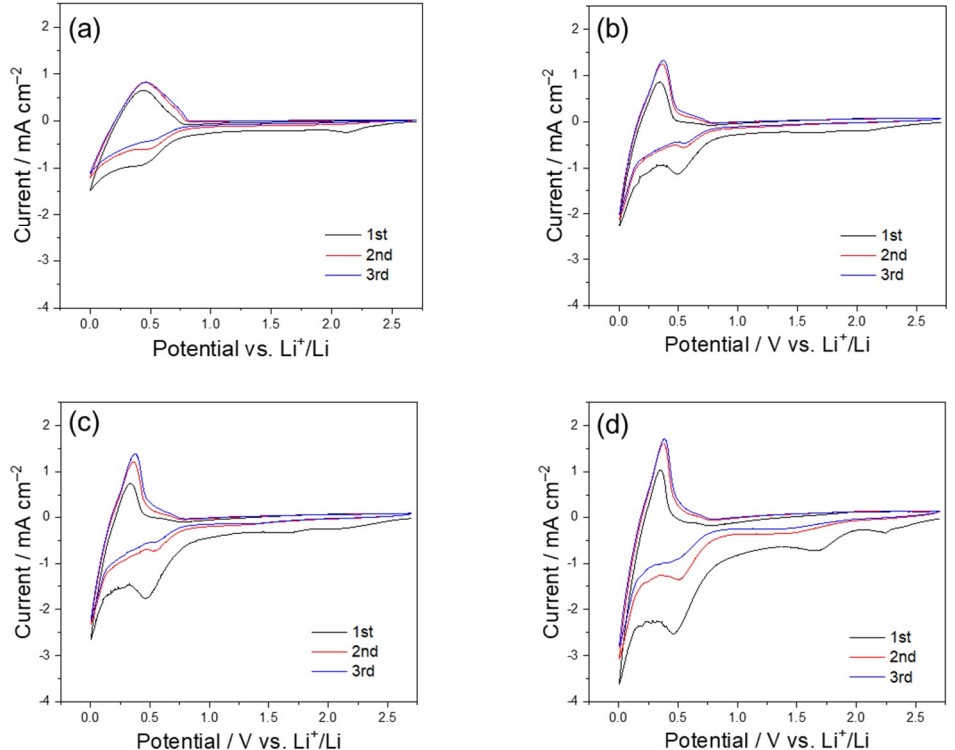

**Figure 1.** Cyclic voltammograms of graphite negative electrode in (**a**) PLD310, (**b**) PLD621, (**c**) PLD311, and (**d**) PLD312. Scan rate: 0.5 mV s$^{-1}$.

Figure 2 shows the charge/discharge properties of graphite in the concentrated PC-based electrolyte solutions with and without DCE. The potential plateau at about 0.5 V attributed to electrolyte decomposition was confirmed during the first cycle in the concentrated PC-based electrolyte solution (Figure 2a). In the subsequent cycles, this plateau disappeared, and reversible intercalation and

de-intercalation occurred at the graphite negative electrode. The effective SEI formation suppressed further decomposition in the concentrated PC-based electrolyte solution.

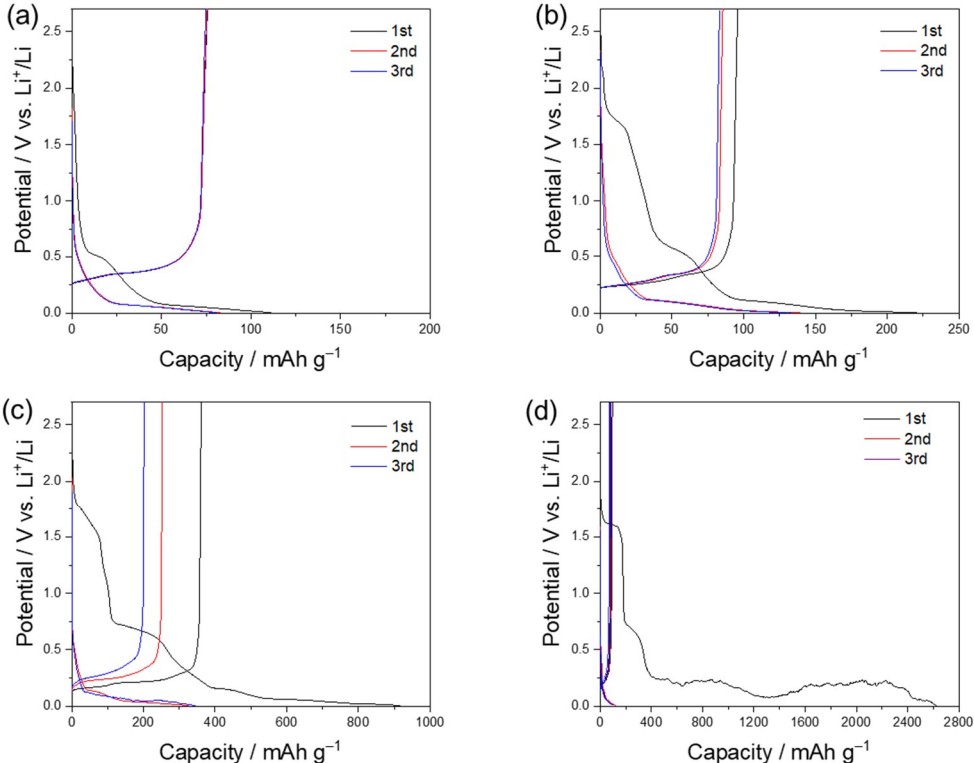

**Figure 2.** Charge and discharge curves of graphite negative electrode in (**a**) PLD310, (**b**) PLD621, (**c**) PLD311, and (**d**) PLD312. C-rate: 0.5 C.

The reversible capacity was estimated to be 75 mAh g$^{-1}$, which is lower than the theoretical specific capacity of graphite (372 mAh g$^{-1}$) because of the low ionic conductivity. Upon DCE addition in the concentrated PC-based electrolyte solution (molar ratio of PC/DCE = 3:0.5 or 3:1), the reversible capacity increased to about 80 and 200 mAh g$^{-1}$ (Figure 2b,c), indicating a remarkable improvement in the electrochemical performance of the graphite negative electrode. However, a large irreversible capacity was observed during the first cycle in PLD312 (Figure 2d) as well as a lower reversible capacity than those in PLD621 (Figure 2b) and PLD311 (Figure 2c). A new plateau was confirmed at about 1.8 V when DCE was added in the concentrated PC-based electrolyte solution (Figure 2b–d), implying the decomposition of DCE. Furthermore, the reversible capacity gradually decreased with the increasing cycle number in the concentrated PC-based electrolyte solutions containing various amounts of DCE. Although this plateau disappeared after the first cycle, the irreversible capacity increased with the increasing DCE amount during the first cycle. To sum up, the addition of a co-solvent (i.e., DCE) to concentrated PC-based electrolyte solutions helped preserve the solvation structure of the lithium ion, enabling effective SEI formation, and increased the ionic conductivity, but the decomposition of DCE was unavoidable.

Raman spectroscopy was performed to investigate the structures of the electrolyte solutions (Figure 3). The peak at 713 cm$^{-1}$ could be attributed to the symmetric ring deformation in the pure PC solvent, implying that the PC molecules exist freely without interacting with the cations or anions (Figure 3a) [17–20].

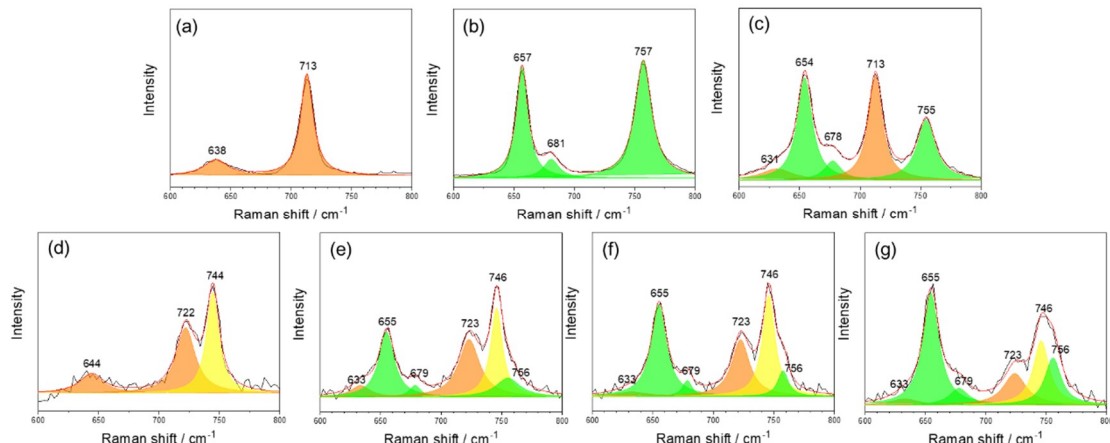

**Figure 3.** Raman spectra of (**a**) PC, (**b**) DCE, (**c**) PC/DCE = 3:1 (molar ratio), (**d**) PC/LiPF$_6$ = 3:1 (molar ratio), (**e**) PC/LiPF$_6$/DCE = 6:2:1 (molar ratio), (**f**) PC/LiPF$_6$/DCE = 3:1:1 (molar ratio), and (**g**) PC/LiPF$_6$/DCE = 3:1:2 (molar ratio).

In pure DCE, the peaks observed at 657, 681, and 757 cm$^{-1}$ were attributed to the Cl–C stretching modes of the gauche conformer with A and B symmetry, and the stretching mode of the trans conformer, respectively (Figure 3b) [21,22]. Significant changes such as peak shifts were not observed, indicating no interaction between PC and DCE in the mixed solvent, as shown in Figure 3c. The peak at 713 cm$^{-1}$ disappeared and shifted to 723 cm$^{-1}$, which means that most PC molecules coordinated to the lithium ions, and the new peak at 745 cm$^{-1}$ corresponding to the PF$_6^-$ anion appeared in the concentrated PC-based electrolyte solution (Figure 3d) [17]. Moreover, the peak at 723 cm$^{-1}$ related to the solvated PC was unchanged in the spectrum of the concentrated PC-based electrolyte solutions containing DCE (Figure 3e–g). No peak shift occurred at 657, 681, and 757 cm$^{-1}$ for DCE as compared with the case of pure DCE. This indicated that the solvation structure of lithium ions in the concentrated electrolyte solution was maintained despite the DCE addition with decreasing viscosity, as shown in Table 1. The effect of DCE addition on the solvation structure was negligible in PLD621, PLD311, and PLD312 because of its low solvation ability. Consequently, an effective SEI was formed and reversible lithium intercalation/de-intercalation took place in the concentrated electrolyte solutions containing DCE. Although the reason for the lowering of the viscosity was not clearly understood, it was confirmed that the electrochemical performance improved due to the increased ionic conductivity in the concentrated electrolyte solution upon the addition of DCE.

## 4. Conclusions

In this study, we investigated the effect of DCE addition on the electrochemical properties of graphite in a concentrated PC-based electrolyte solution. The viscosity drastically decreased in the concentrated electrolyte solution by the addition of DCE, which led to improved ionic conductivity. DCE has low electron-donating ability, so it did not coordinate to lithium ions in the concentrated electrolyte solution. Consequently, reversible intercalation and de-intercalation took place at graphite in the concentrated PC-based electrolyte solution containing DCE, in addition to the enhancement of electrochemical performance. This result indicated that an effective SEI could be formed at graphite despite the decrease in the total molality of LiPF$_6$ upon the addition of DCE. Raman spectra of the electrolyte solution confirmed that the solvation structure of the lithium ions, which enables the formation of an effective SEI, was maintained in the concentrated PC-based electrolyte solution containing DCE. This result suggests that the addition of a co-solvent having low electron-donating ability is an effective method for improving the electrochemical performance in concentrated electrolyte solutions. However, the reversible capacities gradually decreased in the concentrated electrolyte solutions containing DCE, because DCE also decomposed during the initial charge process. Moreover, DCE is unsuitable for application to commercial LIBs because of its high

toxicity and low anodic stability, which need to be resolved for practical LIBs. Despite these bottlenecks, we suggest a new strategy for forming an effective SEI and improving the ionic conductivity in concentrated electrolyte solutions by controlling the solvation structure of lithium ions. In future studies, it will be necessary to clarify the reason for the high ionic conductivity so that suitable co-solvents can be chosen to ensure high safety in practical LIBs. In addition, it will be necessary to prevent the decomposition of the co-solvent while improving the cycling ability, so that the electrochemical performance of LIBs will be substantially improved.

**Author Contributions:** Conceptualization, H.-Y.S.; methodology, M.-H.J. and H.-Y.S.; investigation, M.-H.J.; writing—original draft preparation, H.-Y.S.; writing—review and editing, H.-Y.S. and S.-K.J.; supervision, S.-K.J.

**Funding:** This research was supported by the Basic Science Research Program through the National Research Foundation of Korea (NRF) funded by the Ministry of Science, ICT and Future Planning (NRF-2017R1A2B4010544 and NRF-2019R1A4A1021237). This work was also supported by the Soonchunhyang University Research Fund.

**Conflicts of Interest:** The authors declare no conflict of interest.

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
