# Peer review of "Improving Electrochemical Performance at Graphite Negative Electrodes in Concentrated Electrolyte Solutions by Addition of 1,2-Dichloroethane"

_applsci, doi:10.3390/app9214647_

Round 1
Reviewer 1 Report
Authors studied the effects of addition a co-solvent 1, 2-dichloroethane on the electrochemical performance at graphite negative electrodes.
The paper needs revisions as indicated below:
In material and methods section, authors should mention the chemical companies they purchased their chemicals. And they should explain in more details how they performed the experiments. Section results should be results and discussion and section discussion should change to conclusion. Lines 123-126, sentences need restructure. Use of Fig. or Figure in text should be consistent. Line 140, PF6- Author mentioned the viscosity decreases, they need to show it in a figure, it is only a sentence. In conclusion, author suggested the extension of study, as this study showed the choose of this co-solvent to solve the problems in graphite negative electrodes is effective but they do not have enough characterization support or additional experiments to answer their observation why viscosity decreases and if this co-solvent decomposes, testing other solvents and making comparison to explain the experimental observation and in one concise report is the most effective scientific work.Author Response
Dear Reviewer 1,
We appreciate the time and effort you have dedicated for providing your valuable feedback on our manuscript. We are grateful to you for your insightful comments on our paper. We have been able to incorporate changes to reflect most of the suggestions provided by you. We have highlighted the changes within the manuscript.
Here is a point-by-point response to your comments and concerns.
Authors studied the effects of addition a co-solvent 1, 2-dichloroethane on the electrochemical performance at graphite negative electrodes.
Comment 1: In material and methods section, authors should mention the chemical companies they purchased their chemicals. And they should explain in more details how they performed the experiments.
Response: Thank you for pointing this out. We have added the information about the chemicals in the materials and methods section.
“The concentrated PC-based electrolyte solution was prepared by dissolving LiPF6 (Enchem, battery grade) in PC (molar ratio of PC/Li+ = 3:1). PC (Enchem, battery grade), and DCE (Sigma Aldrich, anhydrous, 99.8 %) mixed electrolyte solutions were prepared by adding DCE as a co-solvent to the concentrated PC-based electrolyte in various molar ratios (PC/Li+/DCE = 6:2:1, 3:1:1, and 3:1:2). Hereafter, the prepared electrolyte solutions will be denoted as PLD310, PLD621, PLD311, and PLD312 according to the molar ratio of PC, Li+, and DCE.
Natural graphite (CGB-10, Nippon Graphite Industries, Ltd.) was used as the working electrode. Natural graphite powder and poly(vinylidene fluoride) (PVdF, Sigma Aldrich, average MW 534,000) binder were mixed in 1-methyl-2-pyrrolidinone (graphite/PVdF = 9:1, wt%) for preparing a slurry, which was then coated on Cu foil and dried in a vacuum oven at 120 °C for 12 h.”
Comment 2: Section results should be results and discussion and section discussion should change to conclusion.
Response: Thank you for your comment. We have changed section titles as follows.
Results → Results and discussion
Discussion → Conclusions
Comment 3: Lines 123-126, sentences need restructure. Use of Fig. or Figure in text should be consistent.
Response: Thank you for your suggestion. We have changed the explanation for clear understanding as follows.
Furthermore, the reversible capacity gradually decreased with increasing cycle number.
→ Furthermore, the reversible capacity gradually decreased with increasing cycle number in the concentrated PC-based electrolyte solutions containing various amounts of DCE.
Comment 4: Line 140, PF6- Author mentioned the viscosity decreases, they need to show it in a figure, it is only a sentence.
Response: Thank you for bringing this to our attention. We agree with your suggestion and have changed the explanation as follows.
This result indicated that the solvation structure of lithium ion in the concentrated electrolyte solution as maintained despite the DCE addition
→ This result indicated that the solvation structure of lithium ions in the concentrated electrolyte solution was maintained despite the DCE addition with decreasing viscosity, as shown in Table 1.
Comment 5: In conclusion, author suggested the extension of study, as this study showed the choose of this co-solvent to solve the problems in graphite negative electrodes is effective but they do not have enough characterization support or additional experiments to answer their observation why viscosity decreases and if this co-solvent decomposes, testing other solvents and making comparison to explain the experimental observation and in one concise report is the most effective scientific work.
Response: Thank you for bringing this to our attention. As suggested, we have attempted the addition of various co-solvents such as benzene, phenol, and cumene for maintaining the solvation structure of lithium ions in a concentrated electrolyte solution. However, we confirmed that the electrolyte solutions could not be mixed into one phase. Hence, we decide to add 1,2-dichloroethane in this study. For the next step, it should be found suitable co-solvents for commercial lithium-ion batteries. Although the physical properties should be improved by using reasonable solvents for practical LIBs, we would like to suggest a new strategy for forming an effective SEI and improving the ionic conductivity in concentrated electrolyte solutions by controlling the solvation structure of lithium ions in this study. Further, we will attempt to investigate more co-solvents for improving safety and stability. We have added the following explanation to the conclusion.
“However, the reversible capacities gradually decreased in the concentrated electrolyte solutions containing DCE, because DCE also decomposed during the initial charge process. Moreover, DCE is unsuitable for application to commercial LIBs because of its high toxicity and low anodic stability, which need to be resolved for practical LIBs. Despite these bottlenecks, we suggest a new strategy for forming an effective SEI and improving the ionic conductivity in concentrated electrolyte solutions by controlling the solvation structure of lithium ions. In future studies, it is necessary to clarify the reason for the high ionic conductivity, so that suitable co-solvents can be chosen for ensuring high safety in practical LIBs. In addition, it is necessary to prevent the decomposition of the co-solvent while improving the cycling ability, so that the electrochemical performance of LIBs is well improved.”

Reviewer 2 Report
In the manuscript “Improving electrochemical performance at graphite negative electrodes in concentrated electrolyte solutions by addition of 1, 2-dichloroethane” the influence of 1,2-DCE on the graphite electrode for Li-ion batteries is discussed.
There are some fundamental lacks in the study which necessitate a broad revision of the manuscript. (Actually, the paper should be rejected completely due to fundamental issues.)
I do not support a publication in its present form due to the following reasons:
Chloro-organic solvents can react in explosion with alkali metals. Thus, the mixture is certainly not suitable for alkali metal batteries (as Li is). Here Li can be present in its metallic form (dendritic) and the temperature in the interior can be increase during usage/cycling. 1,2-DCE is a carcinogenic compound (H 350, Kat. 1B) with a boiling point below 90 °C. This is definitely not suitable to be used in commercial Li-ion cells (safety regulations), especially in high concentrations. It is expected that the anodic stability of 1,2-DCE is not suitable for higher voltages (> 3 V), which restricts the use in Li-ion cells.Nevertheless, the mixtures might be investigated in more basic studies with the intention to proof some mechanistic details.
However, the choice of the mixtures is incomprehensible: It is quite reasonable that ion conductivity and viscosity becomes “better” when the conducting salt is more diluted. Thus it seems more reasonable to fix the concentration of the salt and vary the ratio between PC and DCE. When varying the salt-concentration, a clear salt-to-solvent-mixture-dependency can not be deduced. It can clearly be seen from the CD data that the SEI is not suitable to ensure a reversible cycling. Instead, it is suggested that the irreversible decomposition becomes more and more pronounced when increasing the DCE content (most pronounced in mixture PLD312). Same scale should be used for Fig. 2. Thus it arise the question: Why DCE was used? And what benefit can be deduced? Why the authors use a highly toxic and flammable (fp of 13 °C) compound to enable the use of PC instead of using non-toxic carbonates which can be used immediately without having “propylene carbonate intrinsic” problems? Why non-coordinating less-toxic compounds were not studied, e.g. alkylfluorides, cycloalkanes or others? Certainly, the solubility might be a problem, but then the study should not advertise with Li-ion cells. RAMAN data suggest a non-coordination but they do not prove non-coordinating of DCE. Thus the overall outcome is limited. Charge / discharge should be done at lower C-rate. Then the conductivity becomes less important and a reversible cycling can be identified more easily.
- SI units should be used (not cP).
- The discussion part is more or less a conclusion part.
- no information about the chemicals are presented
Author Response
Dear Reviewer 2,
We appreciate the time and effort you have dedicated for providing your valuable feedback on our manuscript. We are grateful to you for your insightful comments on my paper. We have been able to incorporate changes to reflect most of the suggestions provided by you. We have highlighted the changes within the manuscript.
Here is a point-by-point response to your comments and concerns.
In the manuscript “Improving electrochemical performance at graphite negative electrodes in concentrated electrolyte solutions by addition of 1, 2-dichloroethane” the influence of 1,2-DCE on the graphite electrode for Li-ion batteries is discussed.
There are some fundamental lacks in the study which necessitate a broad revision of the manuscript. (Actually, the paper should be rejected completely due to fundamental issues.)
I do not support a publication in its present form due to the following reasons:
Comment 1: I do not support a publication in its present form due to the following reasons:
Chloro-organic solvents can react in explosion with alkali metals. Thus, the mixture is certainly not suitable for alkali metal batteries (as Li is). Here Li can be present in its metallic form (dendritic) and the temperature in the interior can be increase during usage/cycling. 1,2-DCE is a carcinogenic compound (H 350, Kat. 1B) with a boiling point below 90 °C. This is definitely not suitable to be used in commercial Li-ion cells (safety regulations), especially in high concentrations. It is expected that the anodic stability of 1,2-DCE is not suitable for higher voltages (> 3 V), which restricts the use in Li-ion cells.
Nevertheless, the mixtures might be investigated in more basic studies with the intention to proof some mechanistic details.
However, the choice of the mixtures is incomprehensible: It is quite reasonable that ion conductivity and viscosity becomes “better” when the conducting salt is more diluted. Thus it seems more reasonable to fix the concentration of the salt and vary the ratio between PC and DCE. When varying the salt-concentration, a clear salt-to-solvent-mixture-dependency can not be deduced. It can clearly be seen from the CD data that the SEI is not suitable to ensure a reversible cycling. Instead, it is suggested that the irreversible decomposition becomes more and more pronounced when increasing the DCE content (most pronounced in mixture PLD312). Same scale should be used for Fig. 2. Thus it arise the question: Why DCE was used? And what benefit can be deduced? Why the authors use a highly toxic and flammable (fp of 13 °C) compound to enable the use of PC instead of using non-toxic carbonates which can be used immediately without having “propylene carbonate intrinsic” problems? Why non-coordinating less-toxic compounds were not studied, e.g. alkylfluorides, cycloalkanes or others? Certainly, the solubility might be a problem, but then the study should not advertise with Li-ion cells. RAMAN data suggest a non-coordination but they do not prove non-coordinating of DCE. Thus the overall outcome is limited. Charge / discharge should be done at lower C-rate. Then the conductivity becomes less important and a reversible cycling can be identified more easily.
Response 1: Thank you for your meaningful comments. We agree with this point. We attempted the addition of various co-solvents such as benzene, phenol, and cumene for maintaining the solvation structure of lithium ions in the concentrated electrolyte solution. However, we confirmed that the electrolyte solutions could not be mixed into one phase. Hence, we decide to add 1,2-dichloroethane in this study. As the reviewers’ comment, it should be found suitable co-solvents for commercial lithium-ion batteries. Although the physical properties should be improved by using reasonable solvents for practical LIBs, we would like to suggest a new strategy for forming an effective SEI and improving the ionic conductivity in concentrated electrolyte solutions by controlling the solvation structure of lithium ions in this study. Further, we will attempt to investigate more co-solvents for improving safety and stability. We have added some explanation to the conclusions section as follows.
“However, the reversible capacities gradually decreased in the concentrated electrolyte solutions containing DCE, because DCE also decomposed during the initial charge process. Moreover, DCE is unsuitable for application to commercial LIBs because of its high toxicity and low anodic stability, which need to be resolved for practical LIBs. Despite these bottlenecks, we suggest a new strategy for forming an effective SEI and improving the ionic conductivity in concentrated electrolyte solutions by controlling the solvation structure of lithium ions. In future studies, it is necessary to clarify the reason for the high ionic conductivity, so that suitable co-solvents can be chosen for ensuring high safety in practical LIBs. In addition, it is necessary to prevent the decomposition of the co-solvent while improving the cycling ability, so that the electrochemical performance of LIBs is well improved.”
Response 2: We have changed the scale for Figs. 2a and b, as follows.
"Please find the attached file."
Response 3: We have added an explanation to the manuscript as follows.
“Moreover, the peak at 723 cm–1 related to the solvated PC was unchanged in the spectrum of the concentrated PC-based electrolyte solutions containing DCE (Figs. 3e-g). No peak shift occurred at 657, 681, and 757 cm–1 for DCE as compared with the case of pure DCE. This result indicated that the solvation structure of lithium ions in the concentrated electrolyte solution was maintained despite the DCE addition with decreasing viscosity, as shown in Table 1.”
Response 4: In the results of charge/discharge tests, we want to show a problem or limitation such as low reversible capacity because of low ionic conductivity at relatively high C-rate firstly in the concentrated PC-based electrolyte solution (Fig. 2a). Indeed, the reversible capacity increases at lower C-rate as the reviewers’ comment. So, we thought that it is difficult to investigate different among them in reversible capacity. In addition, we focused on the reversible lithium intercalation and de-intercalation into graphite in the concentrated PC-based electrolyte solutions containing DCE in this study. Because, if the solvation structures changed in the electrolyte solutions containing DCE, it can be expected that no lithium intercalation into graphite (no reversible reaction). However, further, we will try to investigate electrochemical properties at various C-rate and co-solvents for more deep understanding.
Comment 2: SI units should be used (not cP).
Response: Thank you for your comment. We have used the SI unit for viscosity (Pa s), as follows.
|
Electrolyte solution |
Concentration (mol kg–1) |
Molar ratio (PC/LiPF6/DCE) |
Viscosity (Pa s) |
Ionic conductivity (mS cm–1) |
|
PLD310 |
3.27 |
3:1:0 |
0.580 |
0.41 |
|
PLD621 |
2.81 |
6:2:1 |
0.117 |
0.85 |
|
PLD311 |
2.47 |
3:1:1 |
0.059 |
1.31 |
|
PLD312 |
1.98 |
3:1:2 |
0.030 |
2.84 |
Comment 3: The discussion part is more or less a conclusion part.
Response: Thank you for this suggestion. We have changed the section titles as follows.
Results → Results and discussion
Discussion → Conclusions
Comment 4: no information about the chemicals are presented
Response: Thank you for pointing this out. We have added the information about the chemicals in the materials and methods section as follows.
“The concentrated PC-based electrolyte solution was prepared by dissolving LiPF6 (Enchem, battery grade) in PC (molar ratio of PC/Li+ = 3:1). PC (Enchem, battery grade), and DCE (Sigma Aldrich, anhydrous, 99.8 %) mixed electrolyte solutions were prepared by adding DCE as a co-solvent to the concentrated PC-based electrolyte in various molar ratios (PC/Li+/DCE = 6:2:1, 3:1:1, and 3:1:2). Hereafter, the prepared electrolyte solutions will be denoted as PLD310, PLD621, PLD311, and PLD312 according to the molar ratio of PC, Li+, and DCE.
Natural graphite (CGB-10, Nippon Graphite Industries, Ltd.) was used as the working electrode. Natural graphite powder and poly(vinylidene fluoride) (PVdF, Sigma Aldrich, average MW 534,000) binder were mixed in 1-methyl-2-pyrrolidinone (graphite/PVdF = 9:1, wt%) for preparing a slurry, which was then coated on Cu foil and dried in a vacuum oven at 120 °C for 12 h.”

Reviewer 3 Report
The authors present an interesting strategy to improve the graphite performance of concentrated PC based electrolyte solutions. However, some points need to be improved. For example, the common literature knowledge is insufficiently considered. I strongly encourage to consider such literature knowledge in order to increase the impact of this work.
General questions:
What would happen if other (low donating) co solvents would be used, e.g. linear carboantes? I suggest the PC coordination would remain the same with Li+. Halogens e.g. Chlor in 1,2-dichloroethane (DCE), are undesired for application because of chlor oxidation, shutteling (self-discharge) and gaseous Chlor evolution.
How does your electrolyte behave with cathode materials? What is the onset of electrolyte oxidation?
Introduction
“ Electrochemical lithium intercalation into graphite negative electrodes is a significant issue in lithium-ion batteries (LIBs) [1,2].”
Today it is no issue at all. For sure not a significant one. Please use the past tense e.g. “…it used to be a challenge prior the LIB advent in the 1990s” Tarascon and Tarascon et al, solved the electrolyte compatibility with graphite electrolytes by introducing linear carbonates to EC, which were used commercially afterwards.
Please consider literature:
D. Guyomard and J. M. Tarascon, Solid State Ion., 69, 222 (1994). D. Guyomard and J. M. Tarascon, J. Electrochem. Soc., 139, 937 (1992).
“However, salt addition largely increases the viscosity of the electrolyte solution, consequently decreasing the ionic conductivity, and thereby degrading the electrochemical performance of LIBs. Therefore, a method that improves theionic conductivity is needed for enhancing the electrochemical performance of LIBs in concentrated electrolyte solutions.”
It is not only the viscosity, but additionally also the ion pair formation, which decreases the ionic conductivity. Please consider literature:
Kasnatscheew, R. Wagner, M. Winter and I. Cekic-Laskovic, Topics in Current Chemistry, 376, 16 (2018).
Author Response
Dear Reviewer 3,
We appreciate the time and effort you have dedicated for providing your valuable feedback on our manuscript. We are grateful to you for your insightful comments on my paper. We have been able to incorporate changes to reflect most of the suggestions provided by you. We have highlighted the changes within the manuscript.
Here is a point-by-point response to your comments and concerns.
The authors present an interesting strategy to improve the graphite performance of concentrated PC based electrolyte solutions. However, some points need to be improved. For example, the common literature knowledge is insufficiently considered. I strongly encourage to consider such literature knowledge in order to increase the impact of this work.
Comment 1: What would happen if other (low donating) co solvents would be used, e.g. linear carboantes? I suggest the PC coordination would remain the same with Li+. Halogens e.g. Chlor in 1,2-dichloroethane (DCE), are undesired for application because of chlor oxidation, shutteling (self-discharge) and gaseous Chlor evolution. How does your electrolyte behave with cathode materials? What is the onset of electrolyte oxidation?
Response: Thank you for your comment. We agree with this comment. We have attempted the addition of various co-solvents such as benzene, phenol, and cumene for maintaining the solvation structure of lithium ions in concentrated electrolyte solutions. However, we confirmed that the electrolyte solutions could not be mixed into one phase. Hence, we decided to add 1,2-dichloroethane in this study. We did not consider linear carbonates as a co-solvent because dimethyl carbonate and diethyl carbonate participate in the coordination with lithium ions [S-.K. Jeong, et al., Electrochim. Acta, 47, 1975, 2002; Y. Yamada et al., J. Phys. Chem. C, 113, 8948, 2009]. Although the oxidation stability of the electrolyte solution for commercial LIBs should be improved when adding a co-solvent, we would like to show the concept for forming an effective SEI and improving the ionic conductivity in concentrated electrolyte solutions by controlling the solvation structure of lithium ions. Further, we will try to investigate the electrochemical properties in various co-solvents for application to practical LIBs. We have added some explanation in the conclusion section as follows.
“However, the reversible capacities gradually decreased in the concentrated electrolyte solutions containing DCE, because DCE also decomposed during the initial charge process. Moreover, DCE is unsuitable for application to commercial LIBs because of its high toxicity and low anodic stability, which need to be resolved for practical LIBs. Despite these bottlenecks, we suggest a new strategy for forming an effective SEI and improving the ionic conductivity in concentrated electrolyte solutions by controlling the solvation structure of lithium ions. In future studies, it is necessary to clarify the reason for the high ionic conductivity, so that suitable co-solvents can be chosen for ensuring high safety in practical LIBs. In addition, it is necessary to prevent the decomposition of the co-solvent while improving the cycling ability, so that the electrochemical performance of LIBs is well improved.”
Comment 2: Introduction
“Electrochemical lithium intercalation into graphite negative electrodes is a significant issue in lithium-ion batteries (LIBs) [1,2].”
Today it is no issue at all. For sure not a significant one. Please use the past tense e.g. “…it used to be a challenge prior the LIB advent in the 1990s” Tarascon and Tarascon et al, solved the electrolyte compatibility with graphite electrolytes by introducing linear carbonates to EC, which were used commercially afterwards.
Please consider literature:
Guyomard and J. M. Tarascon, Solid State Ion., 69, 222 (1994). D. Guyomard and J. M. Tarascon, J. Electrochem. Soc., 139, 937 (1992).
Response: Thank you for your suggestion. We agree with this comment. Therefore, we have changed the explanation and added the references in the introduction section as follows.
“Electrochemical lithium intercalation into graphite negative electrodes is a significant issue from the viewpoint of diversity of electrolyte solutions applicable to lithium-ion batteries (LIBs) [1-4].”
“3. Guyomard D.; Tarascon J.M. The carbon/Li1+xMn2O4 system. Solid State Ion. 1994, 69, 222-237.”
“4. Guyomard D.: Tarascon J.M. Li metal-free rechargeable LiMn2O4/carbon cells: their understanding and optimization. J. Electrochem. Soc. 1992, 139, 937-948.”
Comment 3: “However, salt addition largely increases the viscosity of the electrolyte solution, consequently decreasing the ionic conductivity, and thereby degrading the electrochemical performance of LIBs. Therefore, a method that improves the ionic conductivity is needed for enhancing the electrochemical performance of LIBs in concentrated electrolyte solutions.”
It is not only the viscosity, but additionally also the ion pair formation, which decreases the ionic conductivity. Please consider literature:
Kasnatscheew, R. Wagner, M. Winter and I. Cekic-Laskovic, Topics in Current Chemistry, 376, 16 (2018).
Response: Thank you for this suggestion. We have added the explanation and reference to the introduction section, as follows.
“However, salt addition largely increases the viscosity of the electrolyte solution resulting from forming ion pairs and aggregates, consequently decreasing the ionic conductivity, and thereby degrading the electrochemical performance of LIBs [15].”
“15. Kasnatscheew J.; Wagner R.; Winter M.; Cekic-Laskovic I. Interfaces and materials in lithium ion batteries: challenges for theoretical electrochemistry. Top. Curr. Chem. 2018, 376, 16.”
Round 2
Reviewer 1 Report
This revised version is accepted to be published.
Reviewer 3 Report
The manuscript is revised accodingly.